# Encapsulation in Alginates Hydrogels and Controlled Release: An Overview

**DOI:** 10.3390/molecules29112515

**Published:** 2024-05-26

**Authors:** Camille Colin, Emma Akpo, Aurélie Perrin, David Cornu, Julien Cambedouzou

**Affiliations:** IEM, University Montpellier, ENSCM, CNRS, F-34095 Montpellier, France

**Keywords:** hydrogels, alginates, encapsulation, controlled release

## Abstract

This review aims to gather the current state of the art on the encapsulation methods using alginate as the main polymeric material in order to produce hydrogels ranging from the microscopic to macroscopic sizes. The use of alginates as an encapsulation material is of growing interest, as it is fully bio-based, bio-compatible and bio-degradable. The field of application of alginate encapsulation is also extremely broad, and there is no doubt it will become even broader in the near future considering the societal demand for sustainable materials in technological applications. In this review, alginate’s main properties and gelification mechanisms, as well as some factors influencing this mechanism, such as the nature of the reticulation cations, are first investigated. Then, the capacity of alginate gels to release matter in a controlled way, from small molecules to micrometric compounds, is reported and discussed. The existing techniques used to produce alginates beads, from the laboratory scale to the industrial one, are further described, with a consideration of the pros and cons with each techniques. Finally, two examples of applications of alginate materials are highlighted as representative case studies.

## 1. Introduction

Alginates were discovered and patented by the British chemist and pharmacist Edward C. C. Stanford (1837–1899) in 1881. It is a well-known family of anionic bio-polymer typically obtained from brown seaweed (around 40% of the dry matter) [1,2] and also from some bacteria, such as *Pseudomonas aeruginosa*. Until now, they have been deeply investigated, as an alginate is a cheap and non-toxic polymer with very interesting gelation capabilities when put in the presence of divalent cations such as ${\mathrm{Ca}}^{2+}$. The worldwide annual production of alginate is estimated at 30,000 metric tons [3], showing that alginates not only stand as a research subject but also present industrial interest.

The encapsulation mechanism will be detailed later, but it is fair to say that it is quite an easy process that can be performed at room temperature in physiological pH conditions. The alginate capsules will then provide a safe environment for many kinds of bio-active/living species [4]. As a result of their properties, alginates seem to be the perfect materials for the immobilization or the encapsulation of various active compounds from bio-molecules [5] and cells [6] to proteins or even microorganisms. The active compound will be trapped in the alginate gel matrix and protected from the surrounding environment while being slowly released from the gel. Another benefit of using alginates as an encapsulation matrix is the control the capsule can give over the release of the entrapped active compound. With the encapsulation, the release can be spread over time, which can be crucial for some application in drug delivery, for example [7,8,9,10].

The encapsulation of active compounds using alginates is used in many areas such as medicine and human health [4,11,12,13,14,15,16], the food industry [17,18], crop protection, and biocontrol [19,20,21,22,23,24,25].

This review will first go through the properties of alginates from their chemical structure to the macroscopic properties of the hydrogel. Then, the controlled release of active compounds will be detailed as well as the main production techniques for alginates hydrogels. Finally, two examples will be deepened concerning encapsulation in alginates coating.

## 2. Alginate Properties

### 2.1. Chemical Structure

Alginates are natural polysaccharides composed of two monomers: the α-l-guluronic acid (G) and the α-d-mannuronic acid (M). What we called an alginate is then a 1→4 linked co-polymer (Figure 1) of these two monomers in various proportions depending on what type of brown seaweed it comes from [11,26,27]. The investigation of the structure of the alginate, in particular, the quantification of the M and G distribution, has been conducted using ^1^H NMR [28,29] (Figure 2). What makes alginates so interesting is their capability of forming a gel by reacting with divalent cationic species. Some work has been carried out in the past thirty years to find a link between the chemical structure of the alginates and this particular ionic reaction [30,31]. These previous studies all concluded that mastering the alginate composition is crucial to understand the interaction between the bio-polymer and the cations.

The main result of the study of the alginate structure is that divalent cations will bind preferentially with the G-blocks. Therefore, the interaction between alginate and divalent cations can be summarized as the interaction between G-blocks and the cations, M-blocks and MG-blocks playing an insignificant role in the binding compared to G-blocks. More precisely, the affinity of the ion for the polymer follows this order: MG-blocks < MM-blocks < GG-blocks [12,31]. It also means that a stiffer material in terms of the Young modulus [31] can be obtained by using an alginate rich in G-block (Figure 3), as the obtained hydrogel will be composed of a large amount of a strongly reticulated area thanks to the high content of G-blocks. Then, it appears that not only is the composition (i.e., the M/G ratio) important, but also the length of the G-block and the molecular weight are other critical factors affecting the physical behavior of the gel [11].

### 2.2. Bio-Compatibility and Bio-Degradability

Many studies on the bio-compatibility and bio-degradability of alginate-based materials have been performed, as those properties are essential for the vast majority of alginates applications [32,33,34,35]. Alginate is a U.S. FDA approved material with already a lot of applications in food and human health fields. Amirian et al. performed histological studies to investigate if the in vivo introduction of alginate hydrogel would provoke any undesired reactions such as inflammation or fibrosis. Results showed that no such reactions were observed after the introduction of alginate micro-beads, meaning alginates are safe to use and will not trigger any dangerous response from the host. The intestinal degradation of alginate has also been investigated [34]. This degradation pathway is of prime importance to better understand the becoming of orally taken alginates. The degradation was studied over a 74 day period with pig intestine as models for human. Alginates turned out to have limited digestibility with an important difference between mannuronic acid (M) and guluronnic acid (G). In fact, M segments were found to be two to three times more digestible than G segments, meaning that digestible alginates could be obtained by using polymers rich in M blocks. In case of intravenous administration, special care should be put into the purification of alginates. Non-purified commercial alginates could contain impurities that may trigger an immune response in the host body or even show some toxicity. As a result of all those studies, alginates are considered bio-compatible and bio-degradable, thus making them suitable for many clinical applications.

### 2.3. Hydrogel Formation Mechanism

As said earlier, alginates are able to form a hydrogel in the presence of divalent cations. The main applications of this gels are in bio-medicine, drug delivery [11], and biocontrol for crop protection [20,36]. A hydrogel is a three-dimensional network of polymeric structure with a high water content entrapped in the network. Like the vast majority of hydrogels, the alginate hydrogel is bio-compatible, which is a very important asset for its application in health or crop protection. Several structures can be obtained depending on the method of production (Figure 4). Alginates can also be co-formulated with another bio-polymer (such as chitosan) in order to obtain a multi-layer capsule or to improve the mechanical properties of alginate beads [12].

The most frequent way to form alginates hydrogels is by ionic cross-linking. This technique is rather simple compared to other hydrogel formation methods, such as covalent cross-linking, and only consists of dropping a sodium alginate solution of a known concentration into a bath containing a divalent cation $M^{2+}$. This technique will give a gel with an “egg-box” morphology [31,37] (Figure 5). The general equation for the reaction was described by Hassan et al. in 1992 [38] and is presented in Figure 6.

According to the “egg-box” model [31,37,39], two facing helical stretches of G-sequence will bind a divalent cation in a chelate type of binding. The selectivity of the interaction is guided by the geometrical need of a cavity that only GG-blocks can create (cf. Figure 5). This cavity will allow a very tight entrapment of the divalent cation, which is not possible with the other blocks of the polymer [5,37]. To sum up, each divalent cation will interact with two adjacent G residues as well as with two G residues from the opposing chain, leading to the formation of a hydrogel. This binding mechanism can be described as a cooperative mechanism, where the first cation binding is energetically unfavorable and the following bindings become more and more favorable (zipper mechanism) [12,37]. In the case of ${\mathrm{Ca}}^{2+}$, a minimum of 8 to 20 adjacent G units are needed to create a stable hydrogel [37]. A more detailed blueprint of the cationic binding is presented in Figure 5.

### 2.4. Influence of $M^{2+}$ Nature on the Hydrogel

The nature of the $M^{2+}$ binding cations can be changed, and several cations have already been tested as a cross-linking reagent for alginate hydrogels [13,30,31,38,40,41]. First of all, Smidsrød et al. declared that ${\mathrm{Mg}}^{2+}$ is not suitable for the gelification of alginates [31]. In 1990, Smidsrød et al. proposed the following ranking regarding the strength of the interaction alginate-$M^{2+}$ [42]:

${\mathrm{Pb}}^{2+}>{\mathrm{Cu}}^{2+}={\mathrm{Ba}}^{2+}>{\mathrm{Sr}}^{2+}>{\mathrm{Cd}}^{2+}>{\mathrm{Ca}}^{2+}>{\mathrm{Co}}^{2+}={\mathrm{Ni}}^{2+}={\mathrm{Zn}}^{2+}>{\mathrm{Mn}}^{2+}$.

It should be mentioned that ${\mathrm{Mn}}^{2+}$ has been tested, but the mechanical properties of the obtained gel are so low that it is barely possible to manipulate [40,41]. It has also been demonstrated that the mechanical properties and the Young modulus of the gel are following more or less the same evolution as the strength of the interaction alginate-$M^{2+}$, except for ${\mathrm{Zn}}^{2+}$, which shows one of the highest Young moduli despite its relatively low affinity for the alginates [40]. It has to be mentioned that one can improve the mechanical properties of alginate gel while using ${\mathrm{Ca}}^{2+}$ as binding ions by changing the nature of the counter ion. Cendon et al. [40] indeed showed that by using calcium gluconate instead of calcium chloride, the Young modulus of the gel is significantly increased at fixed calcium and alginate concentrations (from around 800 kPa to 1000 kPa), putting this ionic combination among the best performers, which are Pb, Cd and Zn (all showing Young moduli just above 1000 kPa) [40].

Finally, it has been demonstrated that in the presence of calcium cations, the evolution of the Young modulus *E* follows the equation [41]:(1)$$E=KC_{alg}^{2}$$
where *K* is a constant and $C_{alg}$ represents the alginate concentration. This shows the huge influence of the concentration in sodium alginate on the Young modulus of the obtained hydrogel.

Moreover, Ouwerx et al. [41] showed the existence of a viscosity threshold to the formation of alginate beads. To produce spherical beads, the alginate solution should be, regardless of the cation nature, in the semi-dilute regime: 1 < C[η] < 10, where [η] is the intrinsic viscosity of the solution and C the alginate concentration. The only exception to this observation is for the ${\mathrm{Mn}}^{2+}$ cation. In fact, the affinity for alginates of manganese is so low that the alginate solution needs to be in the concentrated regime C[η] > 10 to allow the formation of spherical beads. The required viscosity is then a function of the cation type, cations with a weaker affinity requiring a higher C[η].

### 2.5. Reaction Kinetics

Considering the kinetics of the reaction, it has been proposed that when a droplet of a sodium alginate sol enters a solution of divalent metal cations, a gel membrane will immediately be formed around the sol droplet [43], ensuring the morphology of the future hydrogel. Then, the cation will diffuse from the electrolyte solution to the core of the droplet creating a gel bead [38,43]. This diffusion mechanism creates inhomogeneity in the final gel with a denser alginate layer on the periphery of the gel due to a highly reticulated alginate compared to the core of the gel [26,43,44,45]. This second step of ionic diffusion is not instantaneous and is a function of the concentration of the sol and the electrolyte. It has been found that the rate of formation of the gel bead is higher in a metal ion electrolyte of lower concentration rather than in higher ones. This is due to the higher mobility of the metal ions in a less concentrated solution, which will foster the diffusion mechanism. A concentrated sodium alginate sol will increase the rate of the gel formation, as the presence of many binding sites will accelerate the formation of the 3D network [38].

It is yet still possible to obtain homogeneous alginate hydrogels. In order to do so, it is necessary to slow down the reaction either by adding a competitor to the binding of ${\mathrm{Ca}}^{2+}$ with the alginate, such as a small amount of ${\mathrm{Na}}^{+}$ or some sulfate buffer that will compete with the carboxylate group of the alginates for the binding of ${\mathrm{Ca}}^{2+}$ [11,44]. It is also possible to slow down the rate of the reaction by controlling the release of ${\mathrm{Ca}}^{2+}$ in the solution. To do so, one should use an inactive source of calcium such as ${\mathrm{CaCO}}_{3}$. Indeed, ${\mathrm{CaCO}}_{3}$ is stable in aqueous solution at pH = 7 and will not generate any calcium cation unless the pH is decreased. Hence, by decreasing the pH using a slowly hydrolysing molecule such as δ-gluconolactone (GDL), it is possible to slowly release calcium cations in the alginate solution [37], thus giving a more precise control of the kinetics of the reaction as described in Figure 7.

### 2.6. Surface Analysis

Another very important point for the study of the encapsulation is the study of the diffusional properties of active species loaded in the alginate gel, i.e., how an active substance will diffuse out of the gel in order to reach its target. However, the textural characterization of such hydrated materials is complicated since most of the techniques—gas adsorption, Hg porosimetry, scanning electronic microscopy (SEM)—require the removal of the water, which will break the hydrogel structure, leading to biased results. Such studies have already been performed using different drying techniques in order to minimize the structure collapse (critical point drying with supercritical ${\mathrm{CO}}_{2}$, freeze-drying, etc.) [41,46] before using SEM characterizations. According to Aston et al. and Koch et al., the cryo-SEM technique allows the most faithful observations of the 3D structure of the hydrogels, keeping in mind that it still does not represent the exact reality due to the small but inevitable damage caused by the freezing process. The samples are plunged in liquid nitrogen or liquid ethane prior to their observations. The results reveal a fairly maintained structure compared to air-dried or vacuum-dried samples [47,48]. However, additional ice formation and cracks due to ice expansion cannot be avoided and unfortunately keep the observations away from what the wet hydrogel structure really is.

A possible way to overcome the issue of the hydrogel drying is to use small-angle X-ray scattering (SAXS). This technique does not require any specific conditions, such as vacuum or dried samples, and can give access, upon proper calibration, to the specific surface area of the studied material. SAXS has been used on alginate beads [43,49] and after applying the formalism of the so-called Porod limit, a specific surface area of 1.3 ± 0.1 $m^{2}\cdot g^{-1}$ has been found in the conditions of the experiment designed by Ghernaouti et al. [49]. Moreover, SAXS measurements revealed a mostly macroporous structure for alginate microbeads.

In the case where the chemical nature of the surface of the beads is to be investigated, for example, when using a coated alginate capsule with poly-L-lysine (PLL) [14], a suitable technique is to use X-ray photoelectron spectroscopy (XPS). It allows for identification at an atomic level of the chemical nature of the molecules composing the surface of the bead. To perform an XPS study, the beads have to be lyophilized. This step will certainly break the 3D network of the hydrogel, but it is not an issue as long as the surface of the beads remains intact.

### 2.7. Optical Properties

One of the advantages of using an alginate hydrogel as an encapsulation matrix is its ability to protect the encapsulated substance from the external environment and especially from the UV, as many molecules, proteins or organism are UV sensitive. Cendon et al. [40] made the first observation on the color of the hydrogel depending on the nature of the divalent cation and its counter anions. To confirm the first optical observation, Cendon’s team performed an UV/visible absorbance study. It is clearly from their results that alginate hydrogels linked by copper, zinc or cobalt cations exhibited the highest UV absorbance. It means that that not only the ionic salt have some interesting UV absorption properties, but also the hydrogel made from these salts conserves these properties.

Only the beads using chloride and lactate as counter-ions show these absorption properties. Using a gluconate will give very transparent beads. Among all the metallic salt used, ${\mathrm{ZnCl}}_{2}$ is responsible for the most intense drop in absorbance when the wavelength reaches the visible area. The properties resulting from the nature of the binding salt may reduce the use of additives needed for the protection of light- and UV-sensitive compounds, such as clay [40]. In fact, the common way to protect light- or UV-sensitive compounds is to add mineral charges. Zang et al. have shown that the survival rate of bacteria was 11% higher in alginate hydrogels filled with 1% kaolin than in regular hydrogels [25].

### 2.8. Swelling and Syneresis

Alginates beads swell under physiological conditions in saline solution depending on the nature of the alginate used [50]. Unfortunately, well-known kinetics theories for chemically cross-linked gels are not applicable in the case of alginate hydrogels [51,52]. Thus, a new comprehensive study has been performed and demonstrated that alginate swelling kinetics follows a second-order equation [51,52]. Then, thanks to its strong ion-binding interactions, G-rich alginates swell less than M-rich alginates, the latter structure being more disposed to deformation [50,53]. It is also known that the type of cross-linking ion may influence the swelling properties of the beads. For instance, beads made with ${\mathrm{Ca}}^{2+}$ swell more compared to the one involving ${\mathrm{Ba}}^{2+}$ [53]. The swelling of alginate beads is not to be neglected in the case of the in vitro use of the beads, as a bead can gain up to 115% of its initial mass just by absorbing the surrounding liquid until it reaches an equilibrium state. This phenomenon lasts around 60 min and is due to the relaxation of the polymeric network because of the osmotic pressure [54]. Finally, the size of the beads is also important regarding the swelling behavior of alginates. Smaller beads will have a higher swelling rate and will reach an apparent equilibrium more rapidly. This is certainly due to the reduction in the diffusion distance in small beads compared to bigger ones.

Syneresis is a macroscopic slow de-swelling of a gel resulting in a loss of liquid. It is a phenomenon commonly observed over time in various systems, particularly in hydrogels, and it often poses a challenge in the manufacturing and conservation of gels [55]. The molecular weight of alginates is believed to be one factor influencing the syneresis. It has been shown that alginates with lower molar mass tend to exhibit a lower syneresis phenomenon [55]. It is also possible to minimize the syneresis by adjusting the pH of the reticulation bath. A pH near the pKa of the two monomers, which is 3.38 for M units and 3.65 for G units, will slow down the syneresis (Bennacef et al. [5]). The weight loss for ten beads has been monitored for different concentrations of calcium or copper cross-linking ions and for different maturation times [56]. The loss of weight due to syneresis seems to occur during the first 15 h; after this period, alginate beads reach a stable weight. This time is independent of the cation nature. Even so, syneresis depends on the calcium concentration but not on the copper one [56]; this is probably due to the higher affinity of alginates for copper than for calcium [13,56].

However, syneresis is not always an issue to get rid of. In fact, controlled syneresis can be used as a way to create cavities in alginates hydrogels. By using core shell polysaccharides, the team of Di Renzo et al. [57] managed to create cavities in the core of the material using controlled syneresis, while the shell was ensuring the structural stability of the whole hydrogel.

## 3. Diffusion Phenomena and Controlled Release

The encapsulation of active substances in any sort of coatings is interesting, as one is able to control the release of the substance, which can be a crucial parameter in many fields of applications.

### 3.1. Small Molecule Release

Studies on the diffusion mechanism of small molecules have been carried out, especially in the case of drug encapsulation. Controlled drug release is indeed crucial, as a too-rapid release could lead to overdose, while excessively slow release may strongly diminish the drug’s efficiency [8,9,10,58,59].

Ferreira et al. studied the controlled release of encapsulated pindolol as a model for intestinal drug-prolonged release [58]. In the case of small molecules like pindolol, retention in the alginate–chitosan matrix is inadequate under acidic conditions, resulting in the molecules being released within half an hour. At pH = 7.4, the retention time is extended; it takes around 6 h for the complete drug release. However, the release of pindolol was significantly sustained when a chemical cross-linking agent such as formaldehyde or glutaraldehyde was added in the formulation, resulting in the formation of a denser matrix than involving only ionic cross-linking agents. The incorporation of a chemical cross-linking agent may be adapted to reach the desired drug dose over the course of a day. The results of Ferreira et al.’s study are displayed in Figure 8.

For example, El-Aassar et al. focused on lecithin anti-cancer encapsulation and controlled release [8]. The researchers discovered that aside from the already well-established pH dependence, the size of the beads played a role in the cumulative release of lecithin. Smaller beads exhibited higher cumulative release due to the reduced diffusion pathway. The influence of the temperature was investigated at 25 °C, 37 °C, and 40 °C. The total release after 10 h was 17.8%, 44.3%, and 53.8%, respectively, possibly because of the higher hydrogel’s density at lower temperature due to there being more hydrogen bonds between the polymer strands. Increasing the temperature broke these hydrogen bonds, thus reducing the matrix density and accelerating lecithin release.

More recently, Ulag et al. examined the encapsulation of Amphotericin-B (AMB) in alginate. In their study, nanoparticles loaded with AMB were prepared by electrospraying a solution of AMB mixed in sodium alginate in a calcium bath [9]. The obtained beads showed an encapsulation efficiency around 35% with a nanometric size. A study of the drug release behavior indicated that AMB was released with a burst during the first 30 min of the study, reaching around 90% of cumulative release followed by a sustained regime lasting for another two and a half hours and finally reaching almost 100% cumulative release.

To enhance the retention of small drugs molecules within the hydrogel network, Anal et Stevens proposed a solution in 2005. They examined the encapsulation of ampicillin in alginate–chitosan beads [59]. Initial ampicillin retention in a pure alginate matrix was low, with 70% diffusion after 4 h and minimal entrapment (around 15%). These poor results could be improved by reinforcing alginate beads with chitosan, resulting in a significant increase in ampicillin’s entrapment rate and retention time. The best results were obtained 281 with multi-layer alginate–chitosan beads (release < 15% after 4 h). Hydrogels cross-linked with tripolyphosphate (TPP) also demonstrated better ampicillin retention compared to regular calcium-cross-linked hydrogels.

### 3.2. Release of Proteins and Macro Molecules

In contrast to small molecules, the structure of alginates significantly influences the release of proteins and macro-molecules [60].

Through the study of bovine serum albumin (BSA) diffusion from alginate gels, Martisen et al. among others [44,50,60] have provided insights into the phenomenon.

Firstly, they observed that the diffusion rate decreases with the decreasing content of GG-blocks as well as with the decreasing length of these blocks [44]. They proposed that the presence of more GG-blocks in the alginate gel indicates larger pores [13,60]. They have also drawn the logical conclusion that a hydrogel made from a concentrated alginate sol will have relatively low diffusion abilities. Differences in diffusion rates were found between inhomogeneous and homogeneous alginates. The lower diffusion coefficient for BSA in inhomogeneous alginate is likely due to the the membrane of highly concentrated and reticulated alginates at the interface with the electrolyte solution [44]. The diffusion is then faster in homogeneous alginate gel formed in the presence of sodium ions.

Moreover, considering that alginate is negatively charged, pH plays a crucial role in the diffusion rate of charged products. For BSA, the diffusion rate will increase with the pH, as the protein will become more and more negatively charged [44], and the same goes for all negatively charged proteins or molecules encapsulated in alginates [26].

Santagapita et al. focused on the differences between wet, vacuum-dried and freeze-dried alginates beads in terms of enzymatic release [61]. They observed that enzyme release from vacuum-dried beads was much faster than from freeze-dried ones. This is likely attributed to the larger pores size of vacuum-dried beads and the hydrogel cracking during the brutal drying process. Regarding wet beads, they not only enhanced the pH influence on the releasing properties of alginates hydrogels but also demonstrated that the use of additives, such as trehalose, pectin, and β-cyclodextrin, could alter the bead structure, thereby influencing the diffusion mechanism. While regular beads followed Fick’s equations for diffusion, beads with additives introduced a mechanism that, besides Fickian’s diffusion, included the relaxation of the polymer chains.

Finally, the controlled release of blue dextran encapsulated in alginate beads was investigated as a model for the controlled release of macro-molecular drugs [62]. The authors clearly established the pH influence on the molecule release (only 10% to 30% of blue dextran released at pH = 1.2 after 8 h while 100% at pH = 6.8). Alginate concentration also plays a role, with over-concentrated solutions hindering bead formation due to increased viscosity, and a too-dilute solution causing spherical beads to lose their morphology. Within the optimal concentration range, variations of the alginate concentration are of minor importance for blue dextran release. The same applies to the gelation time, bead size, and calcium chloride concentration. However, the drying time was found to be crucial; the more beads that were dried, the faster that molecules diffused out due to the creation of cracks in the bead structure.

All this deals essentially with the encapsulation of organic compounds, but it is noteworthy that the encapsulation of nano-scale inorganic material has also been explored [63]. Results of the release behavior of such materials closely mirror those detailed in this section.

### 3.3. Release of Micro-Metric Sized Compounds

Encapsulation in alginates is also employed for larger materials, including living bacteria or micro-metric compounds. Due to their size, their particular release mechanism is not the same as that for smaller objects and will be developed in the following section.

The encapsulation and release of the bacteria *Raoultella planticola* were studied by He et al. in 2015. For this type of compound, the release occurred over a much longer time scale compared to the compound described in the previous sections. He et al., among others [22,23,64], found that the release of *Raoultella planticola* could be divided in two regimes. The releasing rate of the bacteria was the fastest during the first three days. After this initial burst, the rate slowed down until the end of the experiment at day 30. The authors also observed that the bacteria was not completely released during the experiment. Bacteria from the outer layers of the beads were more released than those encapsulated in the core of the hydrogel. Moreover, the addition of increasing amounts of bentonite in the beads along with the bacteria, increased the retention time of the bacteria inside the alginate matrix (cf. Figure 9). Bentonite forms strong interactions with sodium alginate, densifying the network and inhibiting the bacterial release. The presence of bentonite also regulated the continuous release and minimized the initial burst. Concerning the kinetics, the authors are showing that the bacterial release could be fitted to a first-order kinetics equation.

Even though the “initial burst” behavior is the most common one for the release of encapsulated bacteria, the opposite phenomenon can also be observed [65,66]. This less common release pattern is characterized by a first period of very slow release (after 10 days, the beads studied by Shcherbakova et al. [65] only released 5–8% of their initial content). The burst occurs after this initial phase; after 20 days, the bacteria were mainly out of the beads. The same holds for the work of Liffourrena and Lucchesi [66], where during the first seven days, the release was the slowest. This release behavior is mainly observed in the case of bio-encapsulation for bio-control applications, and its cause is not very clear in every case. For Shcherbakova et al., this late burst in release is attributed to the increased exudation of plant roots with organic acids, accelerating the degradation of the alginate matrix. Another reason for this late release could be the time-dependent erosion and degradation of the capsules. The unstable nature and the bio-degradability of the capsule will lead to the cracking and fragmentation of the alginate network, increasing the release of bacteria [23].

In their study, in addition to this uncommon release behavior, Liffourrena and Lucchesi also highlighted the role of inorganic perlite additive on the release. Beads formulated with perlite showed a fast and significant diffusion, while with very little or without perlite, the release was not significant, and the retention in the beads was very high. The causes of this observation could be attributed to the interaction between perlite and alginate, creating a less dense matrix [66].

## 4. Production Techniques

### 4.1. Extrusion Technique

The most frequent way to encapsulate active compounds in an alginate bead is by extrusion. The molecule of interest is added in the sodium alginate solution. The beads are obtained by dripping this first solution into a bath of divalent cations in water (Figure 10). This technique is very easy to realize, cheap, and highly reliable. The size of the obtained bead roughly corresponds to the size of the droplet and can be thus modulated with the diameter of the syringe, the falling height of the droplet into the bath, the nature of the cross-linking solution, and the viscosity of both the polymer solution and the cation bath [67]. To conclude on this extrusion method, the technique is very efficient for small productions for research use but is hardly relevant for a large-scale or industrial production, as it has a low scale-up ability.

### 4.2. Emulsion Technique

The emulsion is particularly suitable to obtain hydrogel beads in the case that the encapsulation materials are highly soluble in water as in the case of alginates. The idea of this technique is to create a water-in-oil emulsion, where the aqueous phase is a solution of alginate mixed with the compound to encapsulate. By adding a large amount of oil, it will create the emulsion [19,36,67]. The aqueous droplet will, by internal gelation, form the micro-beads after a washing step (Figure 11). The size of the beads is dependent on the type of emulsion and the agitation speed. In the emulsion technique, alginate is the perfect candidate since it has the required high solubility in water compared to other bio-polymers, and it is also able to emulsify a suspension of oil in water. The advantages of the emulsion are that it is easy to scale up and allows flexible adjustment of the resulting capsules. This technique may have a lot of applications in the food sector [67]. Note that in the case of the encapsulation of hydrophobic compounds, emulsion-based strategies can also be considered but in a much more complicated manner than described here.

### 4.3. Spray Drying and Spray Cooling

Spray drying is commonly used in the food industry. It is based on the atomization of an aqueous or oily solution containing the substance of interest and the encapsulation material through a nozzle in a hot chamber. The solvent is vaporized, leaving microcapsules [36,67,68,69] (cf Figure 12). This technique is very cheap, produces high-quality products, and is highly suitable for industrial production [36,67]. However, due to the high temperature required for the process, spray drying cannot be used if the encapsulated compounds are heat sensitive. Other disadvantages of the spray-drying technique are the restriction in the choice of the encapsulation material and the lack of uniformity in the obtained microcapsules under the form of a dry powder [36,67,68]. To prevent the encapsulated materials from mechanical or oxidative stress, it is also possible to add some protective compounds in the formulation of the beads, such as granular starch, soluble fibers, trehalose, and non-fat milk solids [67].

Spray cooling is very similar to spray drying. A molten matrix with a low melting point containing the active compound is also atomized through a nozzle. This time, it is a cold chamber that is used rather than a hot one in order to solidify the matrix around the active compound. Spray cooling is a less expensive technique than hot spray drying, but for now, it is not as widely used in the industry [67].

### 4.4. Fluid Bed Coating

This technique can be understood as a modified spray drying. The molecules of interest are suspended in air, while the encapsulation material is sprayed onto this suspension. This will create a coating around the core, thus encapsulating the molecule of interest [17,67]. A scheme of the fluid bed technique is given in Figure 13. The main advantage of this technique is that the choice of encapsulation material is broader than in the classical spray-drying approach. The success of the coating is strongly influenced by the thickness of the coating material. In principle, the core is always a solid [17].

### 4.5. Electrospinning

The last technique to be presented is the electrospinning technique. Electrospinning, unlike all the other techniques, is not meant to produce alginate beads but is designed to produce fibers. In order to obtain fibers by electrospinning, a strong electric field is created between the syringe and the collector by imposing a high voltage. This voltage will apply an important force on the polymeric molecules in the syringe, and if the voltage is strong enough, this force can overcome the surface tension of the polymer at the end of the nozzle, thus creating a jet of polymer from the nozzle to the collector [70,71]. This technique is already used to produce nanofibers with many different types of polymers, either natural or synthetic, as it is a very convenient way to form nanofibers, a material with many applications, especially in bio-engineering and wound dressing [72]. However, it is reported in the literature that alginates are not suitable for electrospinning due to their electrical conductivity and high surface tension [73]. To overcome this limitation, alginates can be electrospinned using a carrier polymer, i.e., alginates can be blended with polymers that are easily electrospun. Polyvinyl alcohol (PVA) and polyethylene oxide (PEO) are often used as carrier polymers for alginate electrospinning with promising results [74,75], but other polymers are possible [76,77].

Using a carrier polymer allows the obtention of nanofibers but with a relatively low alginate content. Another approach to overcome this issue is to use co-solvents and surfactants in the formulation along with the polymer carrier [70]. The use of the Triton X-100 surfactant can significantly increase the alginate content in the fiber [75], while using DMSO as a co-solvent can change the viscosity of the alginate solution, making it more suitable for electrospinning [78].

Electrospinning seems to be a possible technique for the formulation of alginate materials, but it is not suitable for pure alginates, and the formulation of the to-be electrospun solution should be elaborated carefully to obtain acceptable results.

To conclude with this section, it should be mentioned that this is not an exhaustive list of all the possible techniques but rather an overview of the most popular ones. One could have, for example, mentioned the possibility of producing alginate microcapsules by electrospraying methods or by using a microfluidics setup [79]. In order to summarize the pros and cons of each presented techniques, Table 1 shows the main conclusions from the reference work of Dong and al. [67].

## 5. Case Studies

In this section, two examples of encapsulation using an alginate hydrogel are detailed in order to give the reader some very concrete ways of using alginate as encapsulation materials in two of the most promising areas of application for alginate encapsulation, i.e., anti-cancer therapy and bio-control.

### 5.1. First Example—In Situ Chemoimmunotherapy Hydrogel Elicits Immunogenic Cell Death and Evokes Efficient Anti-Tumor Immune Response [80]

NB: all the following part will be based on the work of Liu et al. [80].

This first example will illustrate the recent use of alginates hydrogels as a vector for anti-tumor therapy.

In this work, injectable alginate hydrogels are used as drug carriers for the more effective treatment of tumors. Alginates are loaded with paclitaxel, a well-known anti-neoplastic agent, bound with albumin to increase its efficiency. This chemotherapy agent is formulated in the alginate along with an immunostimulating agent (R837), as the coupling of chemotherapy and immunotherapy, also called chemoimmunotherapy, has shown very promising results in activating an anti-tumor immune response while creating immunogenic cell death. The major drawback of this very promising technique is the high toxicity for normal cells of chemotherapy and the tumor immunosuppressive microenvironment, making it impossible to administer without a proper vector.

Liu et al. decided to use injectable alginate hydrogel to overcome this difficulty since the precise injection of a loaded hydrogel directly close to the tumor will reduce the risk of diffusion of the paclitaxel and R837 to the surrounding healthy tissue, while ensuring that the tumor is reached by the treatment. Alginate in situ gelation is ensured by the physiological presence of ${\mathrm{Ca}}^{2+}$ cations at a concentration of $1.8*10^{-3}\;\mathrm{mol}\cdot L^{-1}$. Under these conditions, a sodium alginate solution at 20 $\mathrm{mg}\cdot {\mathrm{mL}}^{-1}$ gels in around 10 s. Thus, Liu’s team formulated their injectable solution by mixing paclitaxel and R837 in a sodium alginate solution at this precise concentration, resulting in a rapid in situ and in vivo gelation after injection.

This newly obtained loaded hydrogel showed a sustained release character compared with the injection of pure active molecules. This slow-release effect might also reduce the damage to surrounding normal cells. Moreover, in vitro and in vivo tests have shown that this injectable hydrogel for chemoimmunotherapy is very efficient in creating an “in situ vaccine” due to the presence of the R837 immunostimulating agent, while having very promising anti-tumor activity thanks to the paclitaxel.

Liu et al.’s results may pave the way for the development of dual anti-cancer therapies since using injectable hydrogels as drug carriers will certainly reduce the treatment’s toxicity to normal cells, while prolonging its anti-tumor effect due to the sustained release induced by the encapsulation, boosting its efficiency due to precise local injection. Alginates are among the best candidates in making injectable hydrogels, as, besides their bio compatibility and bio-degradability, the required calcium cations are physiologically present in a concentration allowing a very rapid in vivo reticulation.

### 5.2. Second Example—Encapsulation of Plant Bio-Control Bacteria in Alginate Beads [24,36]

NB: all the following part will be based on the work of Saberi Riseh et al. [36] and Barrera-Cortes et al. [24].

This second example will illustrate the use of encapsulation in alginate hydrogels for bio-control application, which is subject that is being more and more studied, as it raises many challenges for a more sustainable agriculture.

In both papers, alginate encapsulation is used for bio-control applications. Bio-control can be defined as an alternative way to protect crop plants against threats such as disease, insects, fungus, pests, etc. Unlike regular pesticides or chemicals, bio-control solutions bet on natural mechanisms and interactions between species. In this study, the bio-control agents are bacteria. In fact, the survival of bio-control bacteria without any protection is poor in host plants. Thus, alginate is used as a bio-compatible, bio-based and bio-degradable encapsulation material that protects bacteria and increases their life expectancy in the host plant. These bacteria will then decrease the activity of a pathogen by interacting with it. For example, *Bacillus thuringiensis* (*Bt*) species is commonly used on cultures as bio-control agents [20,24,36]. The principle of encapsulated bio-control agents is described in Figure 14.

In their work, Barrera-Cortes et al. encapsulated *Bacillus thuringiensis* in alginate microcapsules produced using the micro emulsion technique [24]. Micro capsules with an average diameter of 3 μm were successfully obtained. Bio-assays confirmed the high performance and efficiency of the encapsulated bio-control agent against *S. frugierda* larvae (killing around 70% and blocking the development of 25% of the remaining larvae), meaning no denaturation of the bio-pesticide occurred during the encapsulation process. Moreover, micro-beads showed high resistance to UV radiation, meaning the alginate capsule protects the bio-control agent from the environmental conditions, thus increasing its efficiency in the long term since *Bt* without protection is rapidly degraded due to sun radiation.

## 6. Alginate Limitations

Even though this paper highlights the many advantages of using alginates as encapsulation matrices, this material is not always suitable. Alginates are stable around neutral pH but can be easily de-cross-linked in acidic or basic conditions. At around pH 5 and lower, alginate hydrogels shrink due to proton-catalyzed hydrolysis, while at pH 10 and above, alginates rapidly degrade due to β-alkoxy elimination, leading to the hydrogel’s dissolution [81]. Another important factor limiting the use of alginate encapsulation is its very high sensitivity to calcium chelators, such as phosphate, lactate, and citrate. If calcium alginate is put in the presence of those molecules, the calcium cations ensuring the stability of the hydrogel network will be chelated, thus leading to the total dissolution of the network. Then, as alginates are natural bio-polymers, which is considered an advantage for many applications, this natural origin also means an intrinsically variable structure that can influence the properties of the final material. This variation in structure and properties is a serious limitation to the large-scale use of alginates in bio-medical applications, where such variations from one alginates to another can hardly be tolerated [16]. Staying in the field of bio-medical applications, pure alginate hydrogels exhibit very limited cell adhesion properties, which is in an important obstacle for the possible use of alginate hydrogels in tissue engineering. In order to boost the cell adhesiveness of alginates, Sarker et al. have proposed either to blend or to covalently cross-link alginates with gelatin, resulting in a material showing high cell adhesion, spreading, proliferation, and migration [82].

It should also be mentioned that alginates are not the only materials capable of encapsulating compounds of interests for food, bio-medical, or bio-control applications. For example, hyper-cross-linked polymers (HPCs) have been recently developed for bio-medical applications [83,84]. Those polymeric networks are made through a controlled Lewis acid/base interaction of the 1,4-benzenedimethanol, resulting in a swellable, spherical hollow morphology. This specific architecture gives HPCs remarkable adsorption properties, making them a very promising candidate for the adsorption of pollutants, such as bisphenol A or heavy metals. Moreover, if branched with L-borneol, HPCs gain anti-bacterial properties, paving the way for their application in the medical field [83]. If properly formulated, HPCs can then not only serve as drug carriers due to their 3D porous architecture but also as anti-microbial, bio-imaging, and bio-sensing agents [84]. All those properties exceed the ones of alginates, so HPCs could be considered a better solution in many areas. However, their main drawbacks with regards to alginate are their higher cost and their petroleum-sourced origin, limiting their use in bio-control applications, for instance.

Finally, the production capacity of alginates is also a significant obstacle to the full development of alginate-based technologies. Even compared to other natural polysaccharides, alginate production is marginal despite its great accessibility. For example, the annual production of alginate reaches around 30,000 tons/year, while starch annual production hits 10.5 million tons/year just in Europe. Thus, efforts are needed to better develop the alginate industry so its usage can be expanded.

## 7. Conclusions

Alginate encapsulation is a very efficient, easy to set up, and eco-friendly way to protect substances of interest from environmental conditions until they reach their destination. Due to its outstanding properties such as bio-compatibility, bio-degradation, non-toxicity, and its ability to quickly form a hydrogel when put in the presence of divalent cations, alginate can be used in as many areas such as biology, agriculture, food industry and more.

Dealing with alginate properties, we summarized the main mechanisms involved in the formation of hydrogels by ionic cross-linking. However, there are still remaining questions in order to understand how some experimental parameters are interfering with the reactions. We have also detailed the controlled release properties of alginate beads in the case of compounds belonging to different size scales. The list of candidates for encapsulation may also be extended, as for now the majority of the literature is focusing on cells and proteins. This will surely lead to new studies to be performed to understand the specific interactions between alginate gels and their precious guests. Another topic of interest for researchers in a near future will be the improvement of the alginate properties using natural additives. In particular, the positive effect of poly-L-lysine (PLL) is very well documented, but there is a clear lack of knowledge regarding the other possibilities.

Finally, from an industrial point of view, there are many techniques available to produce alginate micro-beads at a large scale. The most used are spray drying and -cooling and fluid bed coating, but more and more techniques are being developed to increase the productivity and better match the requirements of the industry. Overall, there is no doubt that alginates will be of great interest for the industry in the upcoming years as a very effective, cheap, and eco-friendly material.

## Figures and Tables

**Figure 1 molecules-29-02515-f001:**
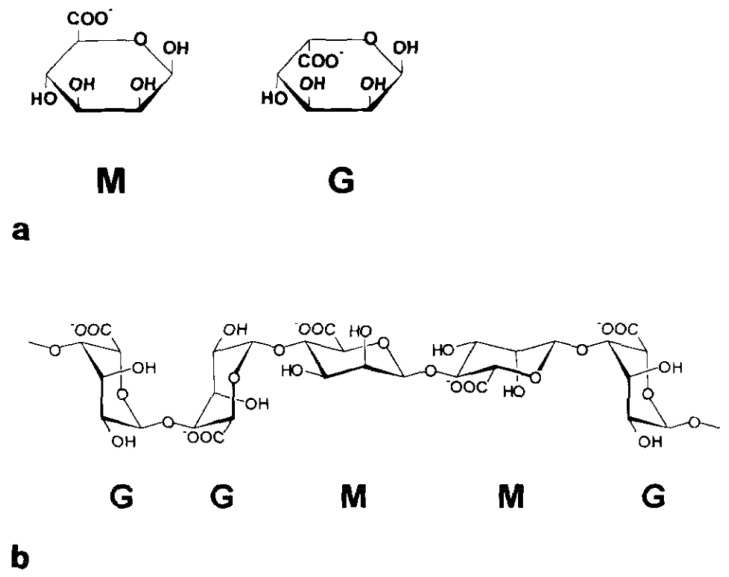
Schematic representation of the α-d-mannuronic acid (M) and α-l-guluronic acid (G) structure (**a**) as well as the polymeric sequence of the alginate with the succession of the M and G blocks (**b**). Reproduced with permission from Thu et al., Colloid and Polymer Science; published by Elsevier, 1996 [26].

**Figure 2 molecules-29-02515-f002:**
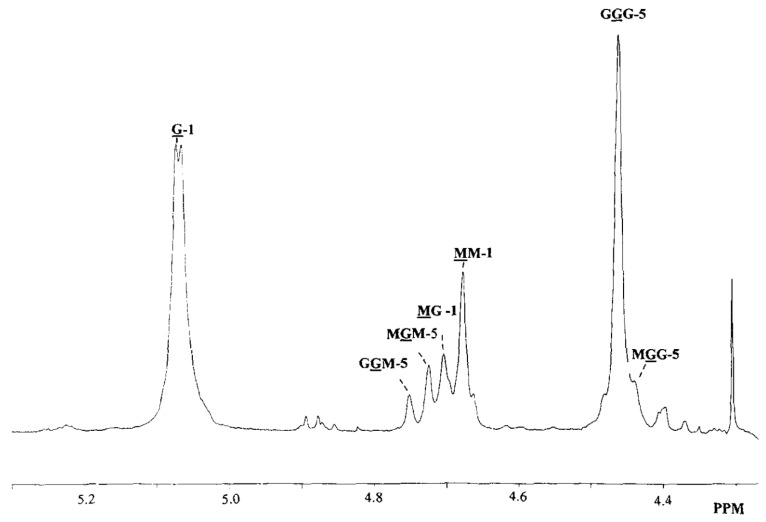
^1^H NMR of the alginate. Reproduced with permission from Thu et al., Colloid and Polymer Science; published by Elsevier, 1996 [26].

**Figure 3 molecules-29-02515-f003:**
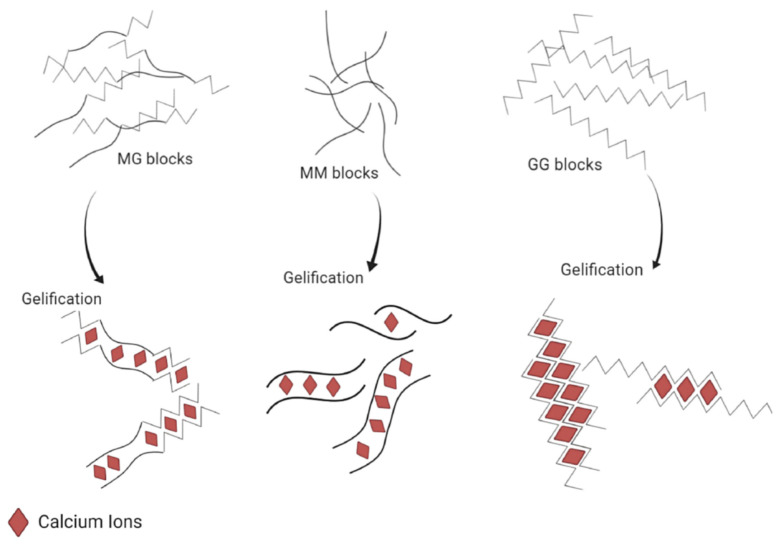
Schematic representation of the different abilities to bind calcium cations depending on the alginate sequence (i.e., MM, MG or GG). Reproduced with permission from Bennacef et al., *Food Hydrocolloids*; published by Elsevier, 2021 [5].

**Figure 4 molecules-29-02515-f004:**
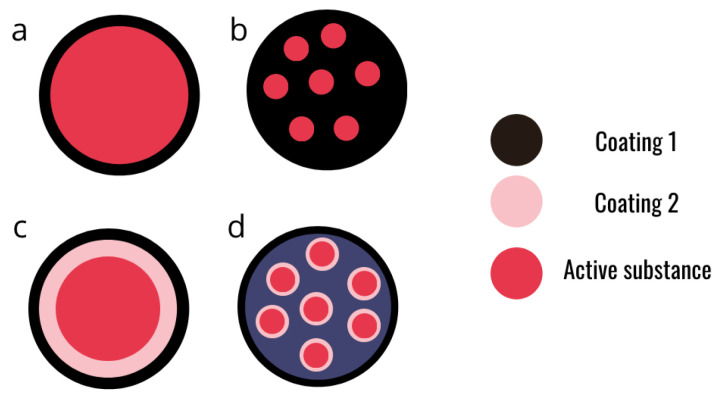
Schematic representation of different designs of alginate microcapsules. (**a**) Simple core-shell micro capsule. (**b**) Matrix micro-capsule. (**c**) Multi-layer micro-capsule. (**d**) Assembly of micro-capsules.

**Figure 5 molecules-29-02515-f005:**
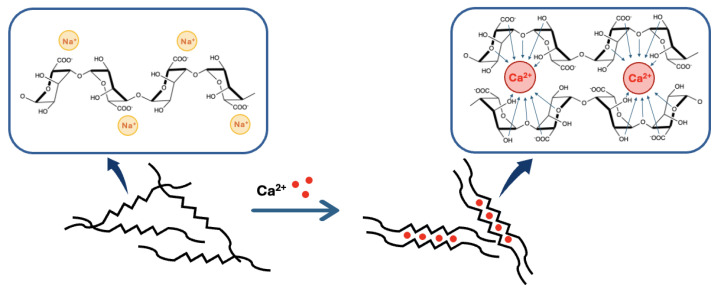
Representation of the binding mechanism of calcium divalent cations by alginate in the so-called “egg-box” model.

**Figure 6 molecules-29-02515-f006:**
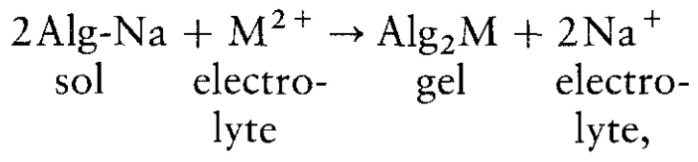
The chemical equation for the ionic gellation of sodium alginate in the presence of divalent cations.

**Figure 7 molecules-29-02515-f007:**
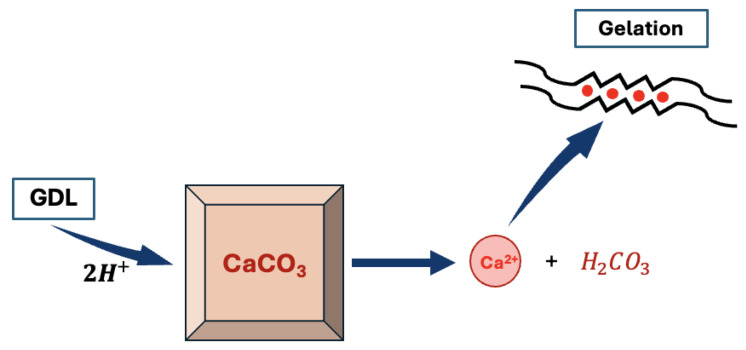
Scheme of the controlled gelification of the alginate using a ${\mathrm{CaCO}}_{3}$/gluconolactone (GDL) system (calcium cations are released in solution after the acidic decomposition of ${\mathrm{CaCO}}_{3}$).

**Figure 8 molecules-29-02515-f008:**
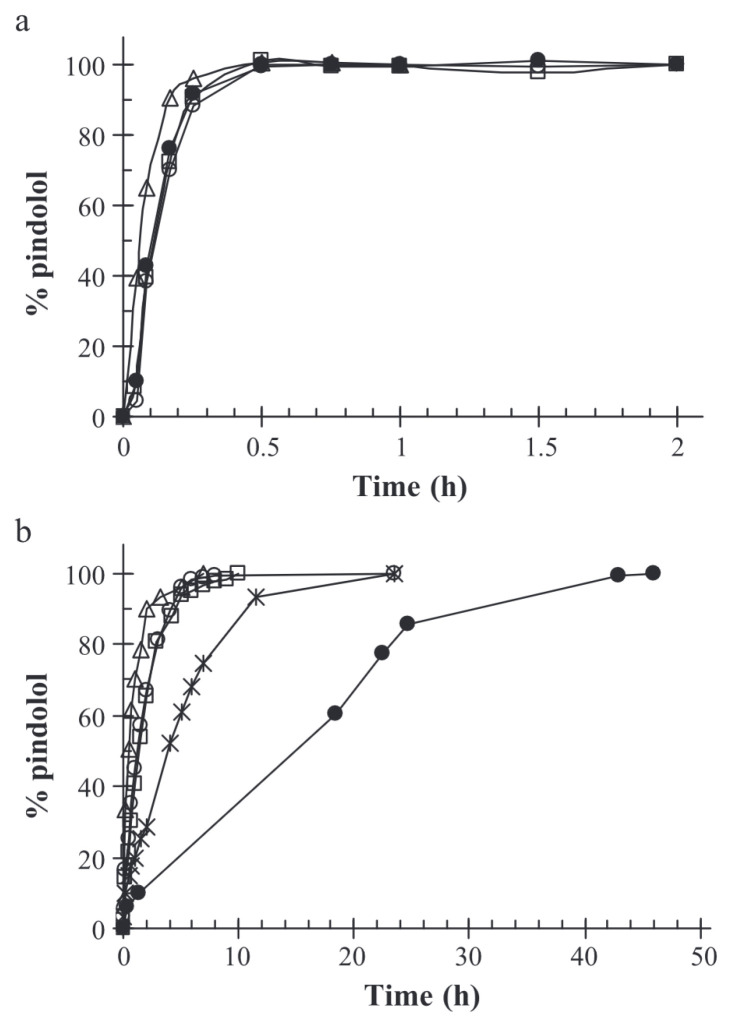
Release curves of pindolol in (**a**) HCl 0.1M, pH = 1,1 and (**b**) phosphate buffer, pH = 7.4. (△) drug-free alginates beads; (□) pindolol-containing beads; (×) pindolol-containing glutaraldehyde cross-linked alginate–gelatine beads; (∘) pindolol-containing alginate–gelatine beads; (•) pindolol-containing formaldehyde cross-linked alginate–gelatine beads. Reproduced with permission from Ferreira et al., Journal of Controlled Release; published by Elsevier, 2004 [58].

**Figure 9 molecules-29-02515-f009:**
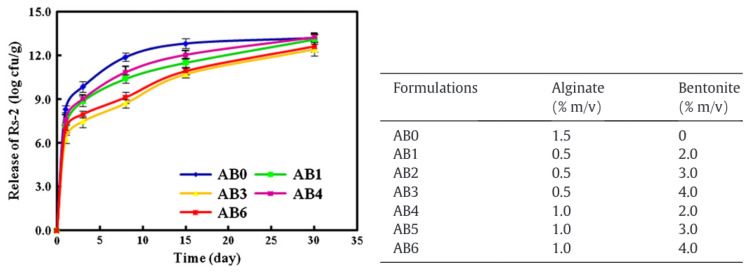
*Raoultella planticola* release depending on the alginates and bentonite concentrations. Reproduced with permission from He et al., Applied Clay Science; published by Elsevier, 2015 [64].

**Figure 10 molecules-29-02515-f010:**
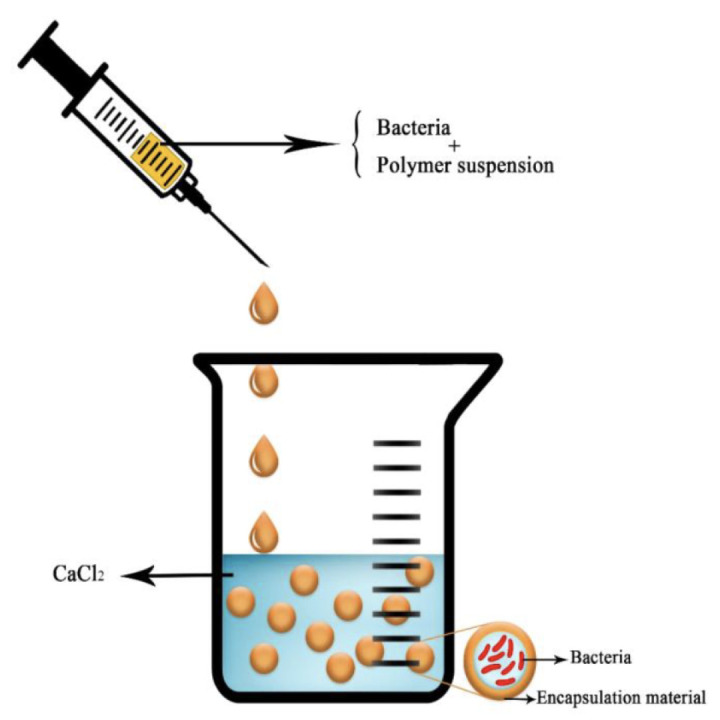
Schematic picture of the extrusion technique with a bacteria as the compound of interest. Reproduced with permission from Saberi Riseh et al., International Journal of Molecular Science; published by MDPI, 2021 [36].

**Figure 11 molecules-29-02515-f011:**
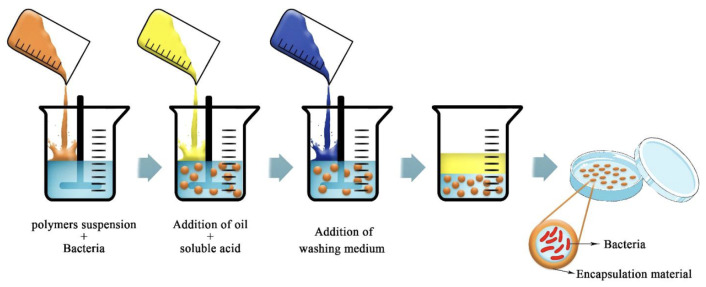
Schematic picture of the emulsion technique for bacterial encapsulation. Reproduced with permission from Saberi Riseh et al., International Journal of Molecular Science; published by MDPI, 2021 [36].

**Figure 12 molecules-29-02515-f012:**
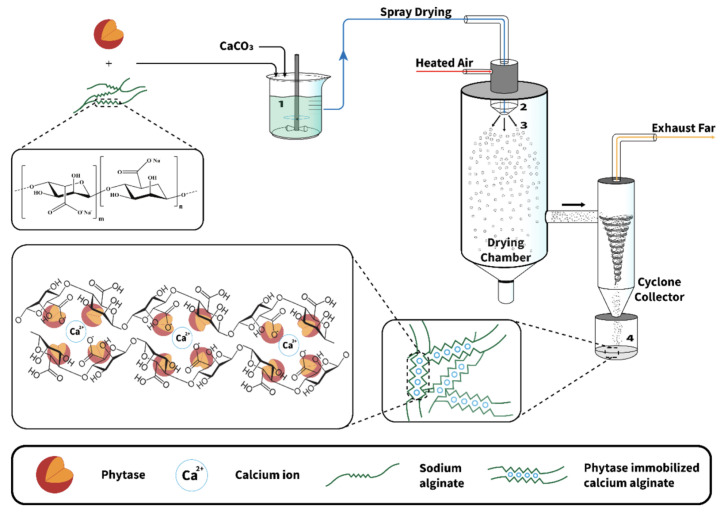
Schematic representation of the overall encapsulation process of enzymes in alginates using spray drying. Reprinted with permission from Weng et al. [68], Copyright 2022 American Chemical Society.

**Figure 13 molecules-29-02515-f013:**
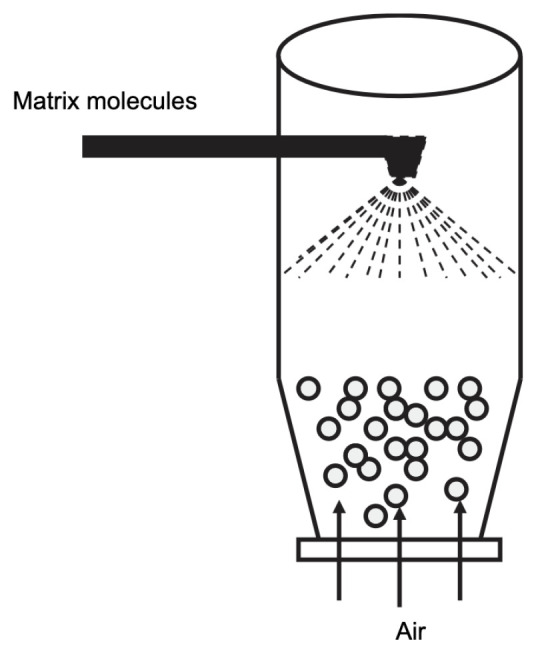
Representation of the fluid bed technology. Reproduced with permission from de Vos et al., International Dairy Journal; published by Elsevier, 2010 [17].

**Figure 14 molecules-29-02515-f014:**
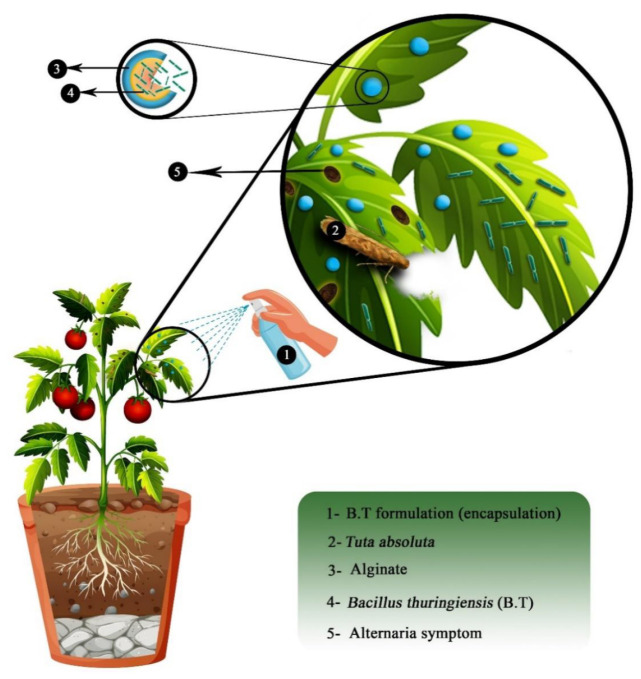
Scheme of the application of alginate encapsulation to protect a bio-control agent. Reproduced with permission from Saberi Riseh et al., International Journal of Molecular Science; published by MDPI, 2021 [36].

**Table 1 molecules-29-02515-t001:** Advantages and disadvantages of encapsulation techniques. Reproduced with permission from Saberi Riseh et al., International Journal of Food and Science Technology; published byWiley, 2013 [67].

**Methods**	**Characteristics**	**Advantages**	**Disadvantages**
Spray drying	Cells encapsulated individually in the drying medium	Many possibilities for coating materials	High temperatures in the process kill many strains
	Medium Cell Load ($10^{10}$–$10^{11}$ CFU $g^{-1})$	Rapid cell release if ingredients dissolve rapidly	
Emulsion	Homogenisation of aqueous and lipid phases	Easy to scale up, flexible adjustment of capsule size.	Emulsifier can be detrimental to viability.
		Lipid-based systems theoretically good for protection against acids and oxygen	High losses in lipid phase
			Lipid core may be unfavourable to long-term stability
Extrusion	Cells blended with various polymers and then extruded.	Many possibilities for coating materials. Particles can be air dried.	Difficulty to be scaled up.
	Low cell load ($10^{9}$–$10^{10}$ CFU $g^{-1}$)	Mild and simple preparation process	
Fluid bed coating	True coating, cell in core powder coat generally lipid based	Easy to be scaled up	Phase separation in beverages if coating is lipid based
	High cell load (>$10^{11}$ CFU $g^{-1}$)	Multiple layers can be added for controlled release or density adjustments	Slow cell release at low temperature

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
