# Peer review of "Encapsulation in Alginates Hydrogels and Controlled Release: An Overview"

_molecules, 2024, doi:10.3390/molecules29112515_

Round 1

Reviewer 1 Report

Comments and Suggestions for Authors

Dear Editor,

Linguistically, the work is well, comprehensibly written. Only there is several spots indicated where corrections are needed.

In terms of content, the work lacks depth, lacks narration, and without that, the work seems like an introduction extended by numerous graphic representations. The manuscript is more reminiscent of a seminar than a review paper. Therefore, I recommend that the paper not be published in the journal Molecules.

Reviewer 2 Report

Comments and Suggestions for Authors

1. Most descriptions are too general, do not provide critical data, and do not reveal possible problems. This is a general review article that somewhat lacks a critical description of the scientific review. Please add one section about the disadvantages of alginate and how to improve that.

2. Some references are too old. Although alginate is an old biomolecule, the review should focus on the recent up-to-date studies. Reducing the old references (older than 10 years) to 20% will be ideal. 

3. The two examples at the end are also too old. Please explain why these examples were chosen and update the article. 

Comments on the Quality of English Language

Minor editing of English language required.

Reviewer 3 Report

Comments and Suggestions for Authors

This review summary the recent approach of the encapsulation in alginates hydrogels and controlled release. The use of alginates as an encapsulation material is of growing interest as it is fully biobased, biocompatible and biodegradable, the alginate properties, diffusion phenomena and controlled release are system reviewed. However, this manuscript so far is not complete, and needs content to be added, I will reconsider my comments after seeing the revised version.

1. The chemical formula is so unclear that it's impossible to read it at all.

2.Electrospinning as an effective molding means needs further clarification

3. Some capitalization issues need attention, and there are many errors in the text.

4. The author needs to give the exact process of cross-linking with diagrams

5.Some of the advantages of this material need to be clarified, such as degradability, stability, etc. Some papers should be cited and compare, such as Materials Today Chemistry 26 (2022): 101252.; J. Polym. Sci. 2024, 62(8), 1517.

Round 2

Reviewer 1 Report

Comments and Suggestions for Authors

Dear Editor,

Still, in my opinion, this paper is not for publication in the journal Molecules because the concept of the paper and the writing style are at the level of a seminar paper. That's why no major or mnor modifications can significantly improve the work.  Here are some concrete examples:

With the encapsulation, the release can be spread over time, which can 33

be crucial for some application in drug delivery for example. [710] 34

-          The paper lacks layering and depth, and since it is an overview paper, here, for example, it is necessary to present in detail what kind of applications it is about.

The investigation of the structure of 48

the alginate, in particular the quantification of the M and G distribution, has been done 49

using 1H NMR [28,29] (Figure 2).
- It is not explained what the assignments 1 and 5 are for.

Many studies on the bio-compatibility and bio-degradability of alginate-based materi- 69

als have been performed as those properties are essential for the vast majority of alginates 70

applications [3235]. - If this is a review article one should be informed if not educated e.g. what kind of applications, insetead of instead of referring the reader to other references

Re-written part:

2.2. Bio-compatibility and biodegradability 68

Many studies on the bio-compatibility and bio-degradability of alginate-based materi- 69

als have been performed as those properties are essential for the vast majority of alginates 70

applications [3235]. Alginate is a U.S. FDA approved material with already a lot of appli- 71

cations in food and human health fields. Amirian et al. performed histological studies to 72

investigate if the in vivo introduction of alginate hydrogel would provoke any undesired 73

reactions such as inflammation or fibrosis. Results showed that no such reactions were 74

observed after the introduction of alginate micro-beads, meaning alginates are safe to use 75

and will not trigger any dangerous response from the host. The intestinal degradation of 76

alginate has also been investigated [34]. This degradation pathway is of prime importance 77

to better understand the becoming of orally taken alginates. The degradation was studied 78

over a 74 day period with pig intestine as models for human. Alginates turned out to 79

have a limited digestibility with an important difference between mannuronic acid (M) and guluronnic acid (G). In fact, M segments were found to be 2 to 3 times more digestible 81

than G segments meaning that digestible alginates could be obtained by using polymers 82

rich in M blocks. In case of intravenous administration, a special care should be put in 83

the purification of alginates. Non-purified commercial alginates could contain impurities 84

that may trigger an immune response in the host body or even show some toxicity. As a 85

result of all those studies, alginates are considered bio-compatible and bio-degradable, thus 86

making them suitable for many clinical applications.

-          The degradation fate in the case of one method of application (alginate micro-beads), and the level of purity in the case of another method of application (intravenous administration) are compared?

-          The conclusions are contradictory!

-          There is no explanation why would intravenous administration require a different level of purity?

-          Finally, it is stated that there is a danger of poisoning, but in the next sentence they are declared as “alginates are considered bio-compatible and bio-degradable, thus 86

making them suitable for many clinical applications.

Authors first discuss figure 6 and later Figure 5?

Reviewer 3 Report

Comments and Suggestions for Authors

Accept in present form